# Implementation of Combined Lifestyle Interventions for Children with Overweight or Obesity: Experiences of Healthcare Professionals in Multiple Communities

**DOI:** 10.3390/ijerph20032156

**Published:** 2023-01-25

**Authors:** Jenneke J. E. H. Saat, Elke Naumann, Gerdine A. J. Fransen, Lieneke Voss, Koos van der Velden, Willem J. J. Assendelft

**Affiliations:** 1Academic Collaborative Center AMPHI, Department of Primary and Community Care, ELG 117, Radboud University Medical Center, P.O. Box 9101, 6500 HB Nijmegen, The Netherlands; 2Research Group Nutrition, Dietetics and Lifestyle, HAN University of Applied Sciences, 6503 GL Nijmegen, The Netherlands; 3Department of Primary and Community Care, ELG 117, Radboud University Medical Center, P.O. Box 9101, 6500 HB Nijmegen, The Netherlands; 4Agrotechnology & Food Sciences Group, Chair Group Nutrition and Disease, Division of Human Nutrition and Health, Wageningen University, 6708 PB Wageningen, The Netherlands

**Keywords:** implementation, combined lifestyle intervention, overweight, obesity, children, multiple case study, network, healthcare professionals

## Abstract

Background: To counteract children with obesity, different protocols for combined lifestyle interventions (CLIs) are implemented by healthcare providers (HCPs). To understand the effects of CLI, we studied the implementation process, facilitators and barriers experienced by HCPs. Methods: A multiple case study design in which community-based CLIs (n = 4), implemented in a total of ten different communities, are conceptualized as a “case”. Qualitative data were collected via group interviews among HCPs (n = 48) regarding their implementation protocol, their network involvement and the adoption of the CLI in a community. Transcripts were coded and analysed using ATLAS.ti. Results: Barriers were the absence of a proper protocol, the low emphasis on the construction of the network and difficulty in embedding the CLI into the community. Funding for these activities was lacking. Facilitating factors were the involvement of a coordinator and to have everyone’s role regarding signalling, diagnosis, guidance and treatment clearly defined and protocolled. HCPs suggested adding certain professions to their team because they lacked expertise in parenting advice and providing mental support to children. Conclusions: Carrying out and adapting the content of the CLI to the community was experienced as easier compared to the management of the organizational aspects of the CLI. For these aspects, separate funding is essential. In the future, mapping the characteristics of a community will help to clarify this influence on the implementation even better.

## 1. Introduction

In 2016, 337 million children between five and nineteen years old were overweight worldwide, 124 million of which were considered obese [1]. Because obesity at a young age affects health later in life (e.g., increased risk for diseases such as type 2 diabetes, cancer, cardiovascular disease [2,3] and a lower quality of life (QOL) [4]), it is important to start prevention and treatment in childhood [5].

In order to counteract obesity in children in an early stage, several countries are implementing multicomponent treatment programs to support children with overweight and obesity in achieving long-term weight reduction [6]. In the Netherlands, such interventions are called combined lifestyle interventions (CLI), in which ‘combined’ resembles the multicomponent aspect of intervention programs [7,8]. The most common components of lifestyle change are: nutrition behaviour through dietary counselling (e.g., advice on a calorie-specific diet), physical activity, and behavioural change (e.g., self-monitoring, setting achievable goals) [7,8] in order to increase the effectiveness of the intervention [8,9,10,11,12,13].

The duration of the intervention is generally two years (one-year treatment and one-year active follow-up (maintenance)). To increase the intervention success, parents are generally involved [6,7,8].

In the Netherlands, preventive screening via weighing and measuring takes place in the youth health care setting, where the youth health care nurse or youth health care physician has the central role in preventive screening for overweight or obesity [8]. Dutch children with overweight or obesity can also be signalled by other health-trained professionals registered in a quality register: a general practitioner (GP), a paediatrician, a dietician or an exercise therapist. In addition to the youth health care nurse or youth health care physician, the GP and paediatrician are allowed to formally diagnose overweight or obesity in children, needed for referring the child to the CLI. Therefore, in case the dietician or exercise therapist signals overweight or obesity, they must refer the child to such a professional.

From age 2 onwards, the body mass index (BMI) (corrected for age and gender) is used as a measure to report the child’s weight and subsequently used for diagnosis. In order to determine whether a child has overweight, the BMI is compared to international BMI cut-off points [8].

Dutch CLIs are public health programmes organized in specific areas in a town (defining the communities). The organization structure of a CLI in practice can be as follows. The initiation is generally local and bottom-up. The initiator of a CLI is usually an entrepreneurial (para)medic. This HCP creates a team to design and implement a CLI. Therefore, the team jointly determines the specific target group of the CLI, for example, children aged 4–8 years. The team as a whole takes responsibility for the implementation and organization of the intervention and results achieved. It may be that one HCP or coordinator of the team is assigned as the driving force. Because the CLI focuses on a combination of three aspects, and because a multidisciplinary approach with multiple health care providers (HCPs) could add to the effectiveness of an intervention [12,13], a CLI is usually carried out by a multidisciplinary team of HCPs. This team is responsible for consulting and supporting the children in their daily life and consists of, for example, a dietitian, a physical therapist and a psychologist. This so-called internal network implements the content of the CLI. In addition, other stakeholders can be involved, forming the so-called external network. These external network stakeholders are involved in the organisational aspects of the CLI, such as signalling, diagnosing, referring, giving direction to the organization, financing and providing support. The external network can consist of, for example, a GP, a youth health doctor or nurse, a social worker, staff of sports clubs, a health broker of the Municipality Health Service and funder(s), such as the municipality and healthcare insurers [7,8]. Healthcare insurance companies in the Netherlands reimburse CLIs as described above since January 2019 for adults, but not (yet) for children. Recently, the Dutch Health Care Institute determined that certain parts (the medical domain) of a CLI for children should be reimbursed by health insurers and other parts (the social domain) not [14]. Conditions defined by health insurers on the reimbursed parts are in development. In addition, a national model for an integrated-systems approach for children with overweight and obesity [15] is currently being rolled out in the Netherlands. Moreover, a new national guideline specifically for children with overweight and obesity [8] (which includes the CLI) has been published recently. However, neither have been implemented nationally yet and are interpreted differently on a regional level. Because of this, and the limited budget, different protocols were and are developed and implemented in practice as local initiatives by entrepreneurial HCPs. Although they are fairly similar, these CLIs differ in, for example, the implementation regarding the content and organisation of the CLI. Additionally, although the same CLI may be implemented in several communities, in principle, differences may exist in the way the CLI is implemented and the way it works in practice in each community.

This variation is due to the fact that the CLI needs to suit local circumstances of the community to reach optimal results [16]. Moreover, the CLI can be adapted by HCPs carrying out the CLI in the community, based on a wide range of implementation theories, strategies, and perspectives, that have been extensively described in experimental studies.

Hence, several protocols for CLIs are implemented in practice in different communities. In order to study and understand the effects of CLIs in the future, insight into what the implementation looks like is needed.

Some research has been conducted on the implementation of obesity prevention programmes, for example, for adults with diabetes [17] or overweight [16,18] and specific for children in school settings [19,20]. As far as we know, implementation processes of CLIs specific for children with overweight and obesity in primary care in the Netherlands have not been studied yet. Studying this is meaningful because of the different funding systems, treatment approaches and training of HCPs [14].

Therefore, the aim of this study is to provide insight into the implementation process, facilitating factors and barriers experienced by HCPs of multiple CLIs in a community setting for children with overweight and obesity in the Netherlands. It is likely that this knowledge contributes to optimizing the implementation process by HCPs and to improving the health-related effects of a CLI. 

## 2. Materials and Methods

### 2.1. Design and Study Population

This study employs a multiple case study design [21] in which each participating CLI (n = 4) is conceptualized as a “case”. The case study approach allows for in-depth multi-facetted explorations of complex issues in their real-life settings [22]. Therefore, case studies are particularly helpful in understanding the internal dynamics of implementation, and including multiple cases capitalizes on organizational variation and permits an examination of how contextual factors influence implementation [22].

During the design and data collection of this study, the four CLIs were already implemented in the region (east of the Netherlands) in four different urban areas. Moreover, these CLIs were implemented by a multidisciplinary network of HCPs who agreed to participate in this study. CLI1 was implemented in four communities, CLI2 was implemented in three communities, CLI3 was implemented in one community and CLI4 was implemented in two communities. One group interview per community was conducted, resulting in a total of ten group interviews, with a range of three to eight participants per interview. The contact person invited the whole team of HCPs to participate per CLI: 18 HCPs working in CLI1, 30 HCPs working in CLI 2, 5 HCPs working in CLI3, and 4 HCPs working in CLI4.

### 2.2. Characteristics of the Cases

Characteristics of the CLIs were obtained using the available documentation of the included CLIs. This documentation varied highly in volume and details.

Table 1 presents the characteristics of the cases and shows that the CLIs are fairly similar.

All four CLIs are group therapies, with group sizes of five to twelve children per group. Moreover, the inclusion criteria of the children are similar: children between 4 and 19 years old with (severe) overweight or obesity (degree I, II and III, with or without co-morbidity and/or risk factors), including their family members. None of the protocols described specific exclusion criteria.

A difference can be seen in the involvement of the disciplines. For example, only CLI1 involved a coordinator (health broker). Another difference was the role of a child health coach (CHC), which was only involved in CLI2. Whenever the CHC assessed the timing was right for the child to receive lifestyle coaching, the lifestyle coaching itself was provided by the CHC and was fully customized.

### 2.3. Data Collection

Qualitative data was collected via group interviews between November 2018 and February 2019. One researcher (J.S.) interviewed the HCPs of the ten teams and with the verbal consent of participants, the interviews were audio-recorded. An interview guide containing themes to be discussed was developed, based on studies that also reported on the implementation of a CLI, but for different ages or settings from ours [23,24,25,26].

During the interviews, we asked what the implementation of the CLI looks like in practice. Because components of the CLI and implementation strategies can be described in a protocol and can be adapted during implementation, we asked about whether a protocol was present, whether it was experienced as complete, whether the protocol was followed (adherence), what the experienced value of a protocol was and whether changes were made over time.

In addition, we asked about the network of professionals, based on the Consolidated Framework for Implementation Research (CFIR) [27]. Because one of the five domains that influences implementation effectiveness is the internal network, we enquired about what the team composition looks like, what the roles of professionals were and how the internal collaboration and collaboration with the external network was experienced.

Moreover, we asked about the experiences regarding adapting and embedding the CLI in relation to community differences because three CLIs were implemented in more than one community. This offered the possibility to make comparisons between cases (the CLIs), but also within the cases themselves (multi-level).

### 2.4. Data Analysis

Group interviews were transcribed non-verbatim. Thereafter, the transcripts were systematically coded and analysed according to the “Framework method” [28] using the ATLAS.ti software (version 9.1.6). Coding of the transcripts was performed by two researchers independently. In a first coding session, the researchers selected text fragments that provided information about the asked topics (top-down or deductive). Thereafter, both researchers examined for text fragments and codes that had not yet been predefined as topics (inductive way or bottom-up method). The results of both coding methods were discussed and the researchers reached an agreement on the text fragments and codes that were ultimately selected. In reporting results, the “COnsolidated criteria for REporting Qualitative research (COREQ)” [29] and standards for reporting qualitative research [30] were used as guidelines.

## 3. Results

### 3.1. Participants

Of the 57 HCPs invited, 48 HCPs participated in a total of ten group interviews (CLI1: 3–5 HCPs, total 17 HCPs, CLI2: 5–9 HCPs, total 23 HCPs, CLI3 and CLI4: 4 HCPs, total 4 HCPs).

HPCs that did not participate mainly provided practical reasons for non-participation, such as insufficient time or own presumption of inadequate knowledge regarding the implementation. An overview of the HCPs that were interviewed per community is provided in Table 2. All the themes mentioned in the data collection process were addressed in the data analysis and no new theme has been added.

### 3.2. Presence and Completeness of a Protocol

For one CLI, there was no protocol. The specificity of the available protocols varied (e.g., logbook with only general goals versus specific protocols per discipline on community level).

Remarkably, nearly all HCPs indicated their protocol was not sufficiently complete to implement the CLI as intended. Insufficient funding and time keep HCPs from evaluating and updating their protocol. According to HCPs, a more complete protocol helps to clarify tasks and responsibilities, thereby promoting collaboration among HCPs. Therefore, a protocol supports implementation particularly at the start of the CLI, with the start of a new group children and when new HCPs join their team.

Despite HCPs of all CLIs indicating that a protocol helps to implement a CLI properly, they also indicated that an incomplete or undetailed protocol did not hinder their implementation of the CLI (“Well, based on experience, you know what to do and that is not necessarily written down on paper.” Dietitian, CLI 4).

### 3.3. Protocol Adherence

Some HCPs mentioned that they deviated purposefully from the protocol based on their professional experience and they used the protocol as a guideline (“You’ve been doing this for so long. At a given moment, you know what to check.” Child health coach, CLI 2; “…if something else is more important for the child than the protocol prescribes.” Physiotherapist, CLI 1). As a result, HCPs designed their own content, which is not predetermined. For example, the theme “snacks” was discussed less in-depth during a group meeting than the protocol prescribed, and materials or methods were not applied as agreed on. A disadvantage of an absent uniform approach according to a protocol is that the effects of a specific CLI cannot be measured properly when HCPs implement the CLI differently.

### 3.4. Internal Network

In short, implementation of the CLI was experienced as difficult if no (protocolled) agreements have been made on everyone’s role in their internal network regarding signalling, diagnosis, guidance and treatment. HCPs indicated that frequent changes in the person representing a discipline hinders proper collaboration. On the other hand, a relatively small number of HCPs (4–6 HCPs) in a stable internal network seems to help them to meet each other frequently. This is needed to engage with the team, find each other quickly in case of problems and to communicate well about experiences and expectations *(*“Clarity regarding roles, tasks and a list of involved professionals of the CLI is necessary to remove a threshold when looking for cooperation.” Dietitian, CLI2). Interestingly, HCPs of one CLI noticed that they were not aware of all the disciplines involved in the implementation of their CLI. For example, the dietitian, psychologist, children’s coach and others introduced themselves to each other upon arrival at the location of the interview, even though the CLI had already been implemented for a year.

### 3.5. Value of Additional Professionals

During the interviews of one CLI, the role of a coordinator was highlighted as very important by the related HCPs. This coordinator worked at the municipality’s Health Services as a health broker and had a professional background in health promotion. According to the HCPs, the health broker was well-aware of what was going on in the community and municipality. The health broker focused on organisational aspects, such as monitoring the implementation and results, providing visibility of the CLI, organising frequent meetings of both the HCPs’ internal and external network, writing annual reports and taking care of financial aspects. With this supporting and coordinating role, the other HCPs’ ability to concentrate on the content of the CLI was facilitated.

In addition, the involvement of an orth pedagogue or a psychologist was highlighted by HCPs of multiple CLIs because of the added value in parenting advice and in providing mental support to children. According to the HCPs, no other profession could perform this task. HCPs of the CLIs who did not involve such a profession had considered involving these, but such a professional was not available in their community or there was a lack of financing for it.

Finally, HCPs also highlighted the so-called child health coach (CHC), which functioned in four communities of one CLI as a central healthcare provider (CHP). Because the CHP functioned as the first HCP, a multidisciplinary trajectory was not initiated directly. This was a different way of working compared to the other studied CLIs, where a child was always treated multidisciplinary by a dietitian and physical therapist from the start of the CLI.

Several HCPs from the internal network of the CLI that involved a CHP mentioned they were not involved in the treatment in order to introduce their expertise into cases that appear to be limitedly complex at first sight, but also into cases that they consider to be directly complex. Several HCPs of the internal network responded in agreement and stakeholders of the external network reacted with wonder. HCPs that referred a child to the CLI, as well as disciplines that were sometimes involved only in complex cases, expressed their doubts about whether the treatment was sufficiently multidisciplinary. Only when the CHP considered that his or her help or expertise was not sufficient, would the CHP seek cooperation with other HCPs, such as a dietitian or physiotherapist. According to the HCPs, this CHP never had contact with, for example, the dietitian in their community. This was difficult for the HCPs to imagine, given the nutritional expertise of the dietitian (requested for treatment in a CLI).

In addition, HCPs expressed their concerns regarding the education of the CHC fulfilling the role of CHP. In primary care in the Netherlands, a CHC is not an established professional. However, in the Netherlands, those who have completed an accredited education in the Netherlands are a “lifestyle coach” or a certified CHP. They were trained in the development and implementation of a program based on evidence-based behavioural change approaches, general coaching strategies and knowledge gained in the course. Therefore, the HCPs of multiple communities of this CLI expressed their concerns about whether an appropriate indication was given to the child, whether other HCPs should be involved in the treatment and if that actually happened.

### 3.6. External Network

First, HCPs indicated that parental involvement was needed to ensure an inflow of sufficient participating children and for completing the CLI. Despite applying specific strategies, the involvement of parents proved difficult and the intrinsic motivation in parents was experienced as low. HCPs were still searching for ways to stimulate and increase parental involvement, such as introducing a financial contribution from parents and minimum attendance registration (a method of external motivation), but this was not considered a desirable solution. (“We often don’t get the money and it takes so much time to go after it. We also wonder whether this contribution will help. It must come from within themselves.” Physiotherapist, CLI1).

Second, sufficient contact and communication with (potential) referrers was needed for an inflow of participating children but was also experienced as difficult by HCPs of all CLIs (“There are children with overweight..., but we do not get them referred from the GPs! Often, I left flyers on the GP’s desk, in waiting rooms of health centres and schools, but it doesn’t help.” Youth health nurse, CLI1; “Even the teacher, who notices there is something unhealthy in the child’s weight, will not start that conversation, while he’s the designated person to do so. They will just not do it.” Youth health nurse, CLI1; “Every professional should take a proactive role in lifestyle support!” Dietitian, CLI1).

Promoting and discussing the CLIs with GPs seems to have a positive contribution to the number of referrals of children. In addition, promotion of the CLI to a wider target group than the GP (for example, other paramedics, school doctors or employees of local sports centres) increased the number of referrals of children. Moreover, it seemed helpful if the CLI was located in a health centre with (potential) referrers and if the GP and involved HCPs of the CLI were part of the same healthcare centre, because in the Netherlands, a health centre receives additional budget contributions for their multidisciplinary activities.

However, promotional activities were an investment carried out in the HCPs’ private time. According to HCPs, there was a lack of time and budget to achieve better communication with (potential) referrers.

Third, the cooperation of a sports partner was described as complicated. This cooperation was considered particularly necessary in communities where only few sport activities were available. The lack of a health-promoting environment and the absence of local activities or sport clubs in the community was mentioned as a barrier by HCPs of all CLIs (“You have to offer a sports activity in the backyard for those children because otherwise, they think it is too far.” Physiotherapist, CLI1). Furthermore, HCPs perceived no connection with parties that offered sports activities and were also wondering if children would join local sports facilities after completion of the CLI.

Fourth, HCPs also experienced insufficient coordination and cooperation between the financers and organizers of the healthcare domain (insurance companies) and the social domain (municipality). For example, it was unclear to HCPs who should finance all the different aspects to implement the CLI properly. In addition, being eligible for contracting with health insurers was experienced as difficult. During the data collection process, differences existed in the way CLIs were financed. All four CLIs were funded by temporary funding sources (grants), such as a (regional or local) innovation funds, the municipality, health insurance company and foundations. However, the available budget to spend differed between the CLIs.

HCPs indicated that an insufficient budget had led to failures in identifying social and medical problems in the child and family early in the contact time of the CLI. Moreover, HCPs stated that they were often forced to treat children in a monodisciplinary manner. This is a result of the lack of budget for time to communicate and collaborate with other HCPs, which hinders a successful integrated approach.

Interestingly, none of the HCPs in the four CLIs experienced a (good) sustainable working relationship with a health insurance company, the municipality or other parties who (partially) reimbursed the CLI (other than annual reporting on the expenditures and results of the CLI).

Lastly, although there was already a protocol in place and experience developed with the implementation of the CLI, in a subsequent community where the same CLI was implemented, the HCPs’ experience of creating an external network was still difficult, as well as reaching agreements on tasks, responsibilities and finances.

### 3.7. Adaptation to Local Circumstances

Although similar CLIs used the same protocol, differences existed in the way CLIs were adapted and embedded in a community. Almost all HPCs of all CLIs experienced difficulties with this process. Interestingly, the HCPs of only one community (CL1) seemed satisfied: their CLI was well protocolled, there was a sustainable internal and external network of HCPs, parents were involved, and there was a sufficient and continuous inclusion of children. A positive experience was that HCPs in this CLI had a good relationship with the community network and joined activities in line with their community (“We need to become more familiar with the health care network. Everyone knows about programmes for adults, but the existence and added value of CLIs for children seems to be new for similar stakeholders.” Physiotherapist, CLI1). HCPs also mentioned their personal contact with staff of sports clubs for promotion of the CLI. However, as mentioned earlier, promotion tasks were an investment of the HCPs’ private time and there was a lack of time and budget to achieve a better and more frequent promotion.

Another team of HCPs noticed that it was helpful to be informed and up to date with related activities in the community, such as theme days, sports days, markets, and open days of schools or healthcare centres, where a social community team has assisted them. In addition, it seems important to be aware of the general lifestyle, concerns and problems of the community.

Compared to implementing the organisational aspects of the CLI, adapting the content of the CLI to the community was easier. This is because the protocol specified what content to transmit and how to do so in the meetings with children or because the content of the CLI was based on experiences of the HCP in applying their own domain-specific professional guidelines.

HCPs indicated that there should also be room for preventing children from having overweight in a more accessible way, for example, by letting children and parents participate in the CLI without taking part in a whole series of meetings (with attendance). In addition, coaching on reaching and maintaining a healthy lifestyle should, according to HCPs, be accessible before being overweight. Therefore, HCPs suggested free walk-in speaking hours in a community centre where parents and children can easily go for a conversation about health and lifestyle. Another suggestion of HCPs was arranging presentations to be continued for citizens of the whole community in order to inform them about a healthy lifestyle.

## 4. Discussion

### 4.1. Main Findings in Comparison to Literature

The aim of this study was to provide insight into the implementation process, facilitating factors and barriers experienced by HCPs of multiple CLIs, for children with overweight or obesity in a community setting in the Netherlands.

Some of our findings resembled recent studies of other CLIs, for example, regarding the limited protocol adherence [16,18,19] and the experienced insufficient contact between HCPs in their network [23,31,32]. As a result, the HCPs of the CLIs we studied indicated a low number of referrals of children with overweight or obesity by the GP. This is in line with studies of van Rinsum [16] and Lau [33], who suggested that intensive contact with and collaborative efforts in the network are crucial for multidisciplinary collaboration and for referring. In addition, van der Heiden et al. [34] concluded recently that GPs have limited awareness regarding CLIs and their content and that even GPs do not know how and to whom they can refer. Their limited belief in the effectiveness of the CLI plays an important role in referring children [34], which could also be the case in the communities we studied.

In addition, similarities were found regarding the inconsistent network. HCPs in our study indicated frequent changes in HCPs as a barrier and they were not aware of all the HCPs and disciplines involved. This is in line with the results of West and Lyubovnikova’s study [35], in which they stated that when disciplines are missing, communication and decision making will be less optimal. This will, as a result, negatively affect patient care and a consistent health care team is, therefore, very important.

Furthermore, studies [16,36] concluded that the help and support from a coordinator or health broker could optimise the programme delivery, multidisciplinary collaboration and the communication of information across the internal and external network, which supports our findings. Moreover, in line with Xyrichis and Lowton’s work [37], HCPs mentioned that it was helpful if HCPs from different communities and also from different CLIs could work together to learn from each other, which could eventually be complemented with national coordination. Similarities were also found regarding financial barriers, which were also identified by HCPs of other CLIs in the Netherlands, such as Cool [16], SLIMmer [17], and Beweegkuur [18]. Funding opportunities and collaborations were, in line with Berendsen et al.’s work [18], found to be important conditions for the sustainability of a CLI and whether communities continued to offer a CLI. The HCPs involved in one CLI mentioned that they thought they were more successful in getting funding allocated than other CLIs, which mainly resulted in more time for multidisciplinary consultation, multidisciplinary collaboration, networking activities and a higher number of involved HCPs.

In contrast, a limited budget results in a low number of children participating in a CLI due to the limited number of CLIs actually implemented in practice. In addition, a low budget leads to insufficient networking, promotion and embedding of the CLI, also resulting in a low number of participating children. Furthermore, it hinders the achievement of a warm transfer to local sports activities in the community. In this way, the limited budget health insurance companies’ and municipalities’ investment in the CLIs plays an important role. Consequently, providing lifestyle support on a daily basis comes down to GPs. Hence, families are now consulting the GP for medical issues, but also for social problems such as loneliness, divorce, financial problems, questions of parenting, etc., which affect their lifestyle [34]. Lifestyle questions do not always have to be answered by a GP. Interestingly, to solve this, HCPs participating in our study indicated that there should also be room for preventing children with overweight in a broader and more accessible sense. When knowledge of medical and social provision in the community is sufficient, combined with a HCPs’ good cooperation, families are supposed to receive the right support and professionals can do what they are trained for.

Finally, regarding this training, HCPs in our study indicated that they were interested in the capacity and the added value of some professions, which Lloyd et al. [20] had already reported and concluded that the personnel used for delivery of the intervention was central in helping the children and consideration of delivery quality and characteristics of personnel was beneficial for a high degree of intervention fidelity.

In addition to similarities with literature, complementary findings and findings specific for CLIs for children were retrieved.

First, insufficient space had been given in the protocols to the child’s social environment (for example, parents and teachers) for stimulating children enrolling and for adoption of the CLI.

Second, HCPs suggest some additional professions for their internal team, such as an orth pedagogue and psychologist.

Third, an important activity related to the successful embedding of the CLI into a community and upscaling is to actively join the community network and organized activities.

Fourth, upscaling a CLI regarding the content seems to be easy, but the maintenance of a stable and sustainable internal network and the building of an external network are difficult. Some protocols of the CLIs we studied did not address this, and funding for this part was lacking.

Fifth, a prerequisite for collaboration into the internal and external network, for embedding the CLI into a community and for upscaling, is to define stakeholders’ roles, tasks and responsibilities regarding the implementation. Even though a national structure of chain care for children with overweight and obesity in the Netherlands has been set up, performance indicators that CLIs must meet are still lacking. It would be useful CLIs developed and implemented as local initiatives by entrepreneurial HCPs can be compared with the indicators and, in the future, financially compensated.

### 4.2. Strengths and Limitations

The first issue concerns the study population. Unfortunately, some of the invited HCPs did not participate. Some HCPs were not present during the interviews because the coordinator of the CLI had not forwarded the invitation. Consequently, we probably missed views of particular stakeholders in the network, such as dietitians, GPs, health insurance companies and/or (parents of) children. As part of the implementation network [8], they could have also had a valuable contribution [27]. In addition, just a small number of GPs has been included in the interviews, most likely because the contact persons did not regard them as part of the team. Given their role in diagnosis and referral to a CLI, we need additional studies into their role perception, ideas, and barriers and facilitators for referral.

Moreover, despite an invitation, some HCPs still did not attend. However, our interviews covered the perspectives of the necessary collaborating professions, which are well-spread across the communities. Therefore, we assume we did not miss important insights or perspectives of specific types of absent HCPs. For instance, a physiotherapist was absent during the group interview in one community, but other physiotherapists were present, or this physiotherapist was present during the group interview in a subsequent community of the specific CLI.

The second issue concerns taking multiple CLIs as cases. While we studied networks that were not implementing the same CLI, the organizations in our sample were comparable in terms of the general objectives and characteristics of the CLI. Based upon these characteristics, programs and practices can be compared in meaningful ways [22].

A weakness is that we did not investigate the characteristics of communities themselves. Mapping the characteristics of a community, for example, socioeconomic status, ethnic background, religion of inhabitants and available facilities and HCPs, provides a better understanding of the fit of the program in the community. This could help to clarify which influence these unique characteristics have on the implementation and implementation setting of a CLI [38]. In contrast, according to Powell et al. [22], we need to shift the focus away from implementing solitary practices towards fostering evidence-based systems and “learning organizations” capable of implementing several CLIs. Therefore, our descriptive data about the implementation processes that networks are currently using is a first step.

Third, in this article, we have not presented details on the different components of the CLIs, partially since there is no clear consensus and taxonomy on these components. Our research group is currently working on a checklist for this, to be used for planning and evaluation of later studies.

## 5. Recommendations

Our findings lead to several recommendations, which can be used in the implementation of CLIs for children with overweight or obesity (see Table 3). These recommendations are based on the results, as well as on an interpretation of the results in combination with the literature.

## 6. Conclusions

The aim of this study was to provide insight into the implementation process, facilitating factors and barriers of a community-based CLI for children with overweight and obesity.

We found that most CLIs did not reach as many children as intended. Implementation issues experienced by the HCPs were the absence of a properly devised protocol, an insufficient internal and external network, and a suboptimal process of embedding the CLI into the community. The most important facilitating factor was to have everyone’s role clearly defined and protocolled regarding signalling, diagnosis, guidance and treatment. To arrange and maintain these organizational aspects properly, separate funding is essential. We think our findings and reflections may help others implement CLIs for children in a more optimal way, including higher work-related satisfaction levels of the involved HCPs. In the future, mapping the characteristics of a community will help to clarify this influence on the implementation even better.

## Figures and Tables

**Table 1 ijerph-20-02156-t001:** Characteristics of the cases. An overview of the combined lifestyle interventions (CLIs) included and their characteristics.

	Treatment Phase	Follow-Up Phase	Individual Meetings	Group Meetings	Involved Disciplines
CLI1	Six months	One year	Treatment phase: every two weeks Follow-up phase: monthly	Treatment phase: every two weeks Follow-up phase: monthly	Sports coach Youth nurse Dietitian Physiotherapist Health broker
CLI2	Six months	Two years	Treatment phase: customized Follow-up phase: customized	Treatment phase: customized Follow-up phase: customized	Child health coach *Optional:* *General practitioner (GP)* *GP manager* *Youth doctor* *Physiotherapist* *Dietitian* *Weight consultant* *Psychologist* *Youth coach* *Coordinators*
CLI3	Six months	One year	Treatment phase: every two weeks Follow-up phase: monthly	Treatment phase: total three meetings Follow-up phase: once	Dietitian Physiotherapist Psychologist
CLI4	Six months	One year	Treatment phase: every two to four weeks Follow-up phase: monthly or once per six weeks	Treatment phase: every two weeks Follow-up phase: monthly	Dietitian Physiotherapist Sports coach Orth pedagogue

**Table 2 ijerph-20-02156-t002:** Interviewed HCPs. An overview of the interviewed HCPs per CLI and community.

**CLI 1**	**Community** **1A**	**Community** **1B**	**Community** **1C**	**Community** **1D**
Total HCPs interviewed n = 17	Sports coach Health broker Youth nurse Dietitian	Health broker Dietitian Physiotherapist	Sports coach Health broker Youth nurse Dietitian Physiotherapist	Sports coach Health broker (coordinator) Youth nurse Dietitian Physiotherapist
**CLI 2**	**Community** **2A**	**Community** **2B**	**Community** **2C**	
Total HCPs interviewed n = 23	Child health coach General practitioner Youth doctor Physiotherapist Youth coach	Child health coach General practitioner Youth doctor Physiotherapist Dietitian Weight consultant Coordinators	Child health coach General practitioner Youth doctor Physiotherapist Dietitian Psychologist General practice manager Coordinators	
**CLI 3**	**Community** **3A**	**Community** **3B**		
Total HCPs interviewed n = 4	Dietitian Physiotherapist Psychologist	Dietitian Physiotherapist Psychologist		
**CLI 4**	**Community** **4**			
Total HCPs interviewed n = 4	Dietitian Physiotherapist Sports coach Orth pedagogue			

**Table 3 ijerph-20-02156-t003:** Recommendations. Overview of recommendations for practice and future research.

For practice:
Create a complete protocol which describes all stakeholders’ roles, tasks and responsibilities. Explore who is responsible for the CLI’s (re)development, implementation, evaluation and financing. Make sure the protocol pays attention to the child’s social environment. Pay attention to building and maintaining networks and their funding. Ensure (new) stakeholders become familiar with the protocol. Ensure that the protocol is a product of all stakeholders, by adapting and improving the protocol regularly. Stimulate protocol adherence. Improve and refresh network skills of stakeholders. Arrange meetings for stakeholders of the internal and external team regularly and discuss responsibilities, tasks, planning and financial aspects. Create a coordinator position (for example, a health broker). Initiate contact with multiple potential referrers or persons who encourage this, such as GPs, professionals responsible for the health of children in schools (school directors/school doctors/school nurses/doctors’ assistants/physical education teachers/health educators), and key figures of sports clubs, and create awareness about the content and added value of the CLI. Organize learning communities of CLIs, where HCPs and coordinators learn from each other’s barriers and successes to eventually be complementary with national coordination of CLIs for children with overweight. Keep on discussing with stakeholders of the healthcare and social domain who are responsible for the initiative’s (re)development, implementation and evaluation. Initiate contact with health insurers and the government to ensure the needed financial resources to implement the CLI. Continuous to the above, add new agreements in the protocol and discuss this regularly with all stakeholders for protocol adherence and impact of the CLI.
For future research:
Study why some communities succeed in initiating and binding stakeholders to their CLI, and other communities fail in this. Further study the success factors that improve the adoption and embedding of the CLI into a community. Examine if there is a most optimal composition of healthcare professionals in the internal CLI team, including the participation of child health coaches, orth pedagogues and psychologists. Study the cooperation between the CLI and local sports partners and clarify if children are being physically active by joining local sports activity’s/facilities after completing the CLI.

## Data Availability

The data presented in this study are available on request from the corresponding author on reasonable request and on signing of a data sharing agreement.

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
