# Peer review of "Implementation of Combined Lifestyle Interventions for Children with Overweight or Obesity: Experiences of Healthcare Professionals in Multiple Communities"

_ijerph, 2023, doi:10.3390/ijerph20032156_

Round 1

Reviewer 1 Report

Overall, use people first language. Starting with the title.

Abstract:

line 18: "several combined lifestyle interventions". I think you mean "different protocols are implemented" 

line 21 (and line 150): "implemented in a total of ten..." 

line 22-23: The phrase "in ten communities among HCPs" is difficult to understand. 'Communities' refers to the people living in a particular area and not the health care setting.

Line 30-31: the conclusion feels contradictory, since the content of the CLI is closely related to the organisation, e.g. involved professions.

Introduction: 

Line 37: add 'worldwide'

Line 55-56: specify 'certain parts' and 'other parts'

Line 46-65: you mention multidisciplinaire teams, multiple HCPs that offer the intervention, entrepreneurial HCPs etc. The organisation structure can be more clarified. Who is responsible for the implementation and organisation and accordingly your target group in this study?

Methods:

Line 93-95: (again) what is your definition of a community? a specific area in town or a population alocated to a multidisciplinary healthcarecenter or ...

Line 107-110: this information belongs to the introduction

The link in Ref 14 is not working

Section 2.2: table 1 suggests that all four CLIs are grouptherapies. This information, as well as the size of the groups cannot be found in text. Same for inclusion criteria of participants

Results:

Table 2 is lacking information on their roles and responsibilities (in the organisation of the intervention) 

Section 3.2: do you have information on the content (/components) of the different CLIs, to get insights in the quality (apart from the fact whether they had a protocol)? In the discussion (line 385) some suggestions were made, but this information was not offered in the results.

Line 220-232: can you offer more insights on which HCPs in those teams had their doubts? Did it involve HCPs that referred their patient to this intervention or disciplines that were sometimes involved only in complex cases, e.g. psychologists? 

Line 272-280: was their a relation between funded activities and differences in protocols? In other words: did the financial recourses drive protocol adjustments? This seems to be suggested in the discussion (line 358-361) but information is lacking here.

Method/result section: I miss information about the communities, e.g.  differences in average social-economic position and rural or urban area. I see that you mention this as a weakness in the discussion. This has large impact on the results and should be made more clear in the summary / conclusion

In the discussion (i.e. line 335 and 367-371) you suggest that the GP has an essential role, however only a little number of GPs have been included in the interviews. Can you explain this?

Line 400-401 is incorrect. This information is included in the new national guideline

Not all of the recommendations can be found in the results. Is this a combination of results from this study and the literature? This needs to be stated more clear.

Reviewer 2 Report

Introduction

Because this is a system that is implemented only in the Netherlands, it would be good if the introduction was more detailed, in the sense that other countries could benefit from these policies. Maybe add a paragraph including some examples of what other European countries are doing similarly. 

Please provide a better and more clear definition for the CLI than “Combined” means that the CLI focuses on a combination of improvement of nutrition behaviour, physical activity and mental health [6, 7] to increase the effectiveness of the intervention [8–11].”

Is the CLI a public health program that includes different types of professionals dealing with those 3 main branches (nutrition, PA, and mental health), as presented in the Methods at Table 1? Make that clearer because to a foreigner of Netherlands policies, it is quite unclear. What are the differences between the 4 CLI?

Please change the phrase in line 60: “And, although the same CLI may be implemented in several communities IN PRINCIPLE, differences may exist in the way the CLI works IN PRACTICE in each community.” You can either keep it like this or rephrase it while keeping the meaning. 

This phrase does not have a verb (lines 63-65): “Moreover, because the CLI can be adapted by relevant HCPs on the basis of a large number of available implementation theories, strategies and perspectives, described in many (experimental) studies.”

I propose: “Moreover, the CLI can be adapted by relevant HCPs based on a wide range of implementation theories, strategies, and perspectives, that have been extensively described in experimental studies.” Or some other rephrase. 

Do not use “So,..” (line 66), use “therefore”, “hence” etc.

Methods

Please provide more details about the communities: rural/urban, how many people approximately live in those communities. 

Please provide more details about what was asked specifically in the group interviews.

Table 1 – please correct the spelling – “customized”, not cOstomized.

Results

The way the results are presented is not appropriate for an Article. Please present the situation described by the HCPs in a more structured way, with tables, figures - tangible results.

For instance, you narrate situations instead of structuring the information regarding each CLP in a comparative way. 

Example: 

During the interviews of one CLI, the important role of a coordinator was highlighted as very important by the related HCPs. This coordinator worked at the municipality. Health Services as a health broker and had a professional background in health promotion. According to the HCPs, the health broker was well aware of what was going on in the community and municipality. The health broker focussed on organisational aspects, like monitoring the implementation and results, providing visibility of the CLI, organising frequent meetings of both the HCP intern and extern network, writing annual reports and taking care of financial aspects. With this supporting and coordinating role, it was facilitated that the other HCPs were able to concentrate on the content of the CLI.”

“HCPs of the CLI that involved a CHP speaked out their doubds, because of their 220 experienced insufficient multidisciplinairy trajectory. Only when the CHP considered that 221 his or her help or expertise was not sufficient, the CHP should sought cooperation with 222 other HCPs, such as a dietitian or physiotherapist. According to the HCPs, this CHP never 223 had contact with, for example, the dietitian in their community. This was difficult for the 224 HCP to imagine, given the nutritional expertise of the dietitian (requested for treatment 225 in a CLI).|

You should have tables or figures highlighting the data observed during interviews.

What works, what does not work. What are the pros, and what are the cons? What proposals did the HCPs have in order to improve their efficiency?

How many HCPs want a certain policy, or a certain addition to the protocols or teams? 

Give numbers, percentages...

Also, I’m not so sure how appropriate the citations from the HCP professionals interviews are for a paper like this.

The way the result section is structured presently is more suitable for discussions than results. Hence, please revise this section accordingly. 

The Discussion, Recommendation, and Conclusions sections are well structured and I have no suggestions for changes.

I would also ask the authors to review the English because there are a lot of grammar and spelling mistakes throughout the manuscript. 

Round 2

Reviewer 1 Report

The manuscript has greatly improved! The authors clearly understood the reviewers' comments and reworded the text accordingly.

However, I still have some questions/ remarks.

Introduction

Line 61-62: "In the Netherlands, such interventions are called Combined Lifestyle Interventions". The name combined lifestyle intervention (CLI) is well known worldwide and used as such in international literature. I saw that reviewer 2 had questions regarding the protocol of the program, but not the name.

Line 63: what do you mean with elsewhere?

Line 64-65: the Dutch healthcare system is quite different as compared to other countries. I.e. preventive screening takes place in youth health care settings.
Moreover, diagnostics (when necessary before starting treatment) should be mentioned shortly. 

Line 77: can you offer an example of specific target groups? 

Line 115: different protocols, instead of "several CLIs"

Line 116: "fairly similar" is not true, I think. Mostly the duration can vary widely

Line 121: what do you mean with "relevant HCPs"?

Methods

Thank you for adding the extra information on group sizes. Can you also add exclusion criteria such as intellectual deficit and severeness of obesity (i.e. comorbidities)? 

And, do I understand it correct that in one group children from the age of 4 years can be treated together with children from the age of 18?

Line 196-199: the Dutch national guideline does not advice the CHC to be the one that offers lifestylecoaching. The CHC is coordinator and organizes the contact between patient and lifestylecoach. So the methodology of CLI2 does not fit the guidelines.

Line 591-594: In the Netherlands, the youth health care nurse or doctor has the central role in screening for overweight or obesity and thereafter refer to a CLI, when appropriate. So it is not surprisingly that GPs were not seen as teammembers. They do not have a central role in child health care, most often.

Table 3, recommendations:

Can you provide references, when appropriate?
